# BioAIEgens derived from rosin: how does molecular motion affect their photophysical processes in solid state?

Xu-Min Cai [1,2,4], Yuting Lin[1,4], Ying Li[2,4], Xinfei Chen[1], Zaiyu Wang [2], Xueqian Zhao[2], Shenlin Huang [1✉], Zheng Zhao[2,3✉] & Ben Zhong Tang [2✉]

The exploration of artificial luminogens with bright emission has been fully developed with the advancement of synthetic chemistry. However, many of them face problems like weakened emission in the aggregated state as well as poor renewability and sustainability. Therefore, the development of renewable and sustainable luminogens with anti-quenching function in the solid state, as well as to unveil the key factors that influence their luminescence behavior become highly significant. Herein, a new class of natural rosin-derived luminogens with aggregation-induced emission property (AIEgens) have been facilely obtained with good biocompatibility and targeted organelle imaging capability as well as photochromic behavior in the solid state. Mechanistic study indicates that the introduction of the alicyclic moiety helps suppress the excited-state molecular motion to enhance the solid-state emission. The current work fundamentally elucidates the role of alicyclic moiety in luminogen design and practically demonstrates a new source to large-scalely obtain biocompatible AIEgens.

[1] Jiangsu Co-Innovation Center of Efficient Processing and Utilization of Forest Rescources, College of Chemical Engineering, Nanjing Forestry University, Nanjing, China. [2] Department of Chemistry, Hong Kong Branch of Chinese National Engineering Research Center for Tissue Restoration and Reconstruction, Institute of Molecular Functional Materials, Division of Life Science and State Key Laboratory of Molecular Neuroscience, The Hong Kong University of Science and Technology, Clear Water Bay, Kowloon, Hong Kong, China. [3] School of Chemistry and Engineering, Southeast University, Nanjing, China. [4]These authors contributed equally: Xu-Min Cai, Yuting Lin, Ying Li. ✉email: shuang@njfu.edu.cn; zhaozheng@seu.edu.cn; tangbenz@ust.hk

Light has played an irreplaceable role in the long history of human development. Besides illumination, light has been applied in many advanced technologies, including optoelectronics, fiber-optic communication as well as optical diagnosis[1–3]. As the source of light, the earliest record of luminescent substances was an infusion of a Mexico wood with blue emission, which can be dated back to the sixteenth century[4–6]. Since then, luminescent matters had attracted people's attention and various luminescent matters such as inorganic stones and natural products or organism were gradually discovered and investigated[7–9]. However, due to the limitations in variety and difficulty in the study of structure–property relationship, the understanding on luminescence phenomenon and the development of luminescent materials is generally lagged. Until the fast development of synthesis technology, facile acquirement of artificial luminescent materials as well as systematical study on the structure–property relationship could be achieved[10,11]. Regretfully, the aggregation-caused quenching (ACQ) phenomenon always occurs, prohibiting its application of luminescent materials in high concentration or aggregated state[12–14]. Therefore, how to address this ACQ problem has become one of the most flourishing research topics in recent decades.

Luminogens with aggregation-induced emission (AIE) has aroused intensive interests since this kind of materials possesses unique characteristic distinguished from that of ACQ ones[15]. Thanks to their aggregate-state emission and aggregation-responsive luminescence, AIE luminogens (AIEgens) have exhibited promising applications in organic optoelectronics, stimuli-responsive materials, chemosensors/biosensors, and luminescent theranostics[16–21]. Thousands of new AIEgens thus have been designed and synthesized, most of which were structurally designed on conjugated moieties. However, many of their preparation procedures require complicated synthesis and purification techniques, making them uneasy to be obtained, let alone their abundant pollution hazards. Alternatively, plant-originated chemicals are green and bioactive compounds that have been widely studied as one of the most interesting nature-based compounds in the advanced (bio)sensing fields because of their numerous inherent and unique characteristics, such as natural availability, renewability, sustainability, biodegradability, and biocompatibility[22]. Hence, AIEgens from natural resources have recently gained researchers' intense attention (Fig. S1)[23–32]. To date, some natural product-based AIEgens including natural polymers (sodium alginate and bovine serum albumin) and small molecules (quercetin, myricetin, and berberine chloride) have been investigated with unique advantages in bioapplications. Due to their advantageous biocompatibility, imaging can be applied either in vitro or in vivo. Nevertheless, in general, the development of these bioproduct-inspired AIEgens (BioAIEgens: AIEgens or AIE-active nano agents obtained from natural resources including natural products or derivatives by modifying natural products)[21] is still in its infant and the limited variety of AIEgens from natural resources hinders their further investigation and development. In consideration of these, BioAIEgens derived from the chemical modification on the abundant and low-cost natural resources might be a good alternate to either pure natural or artificial AIEgens.

Natural rosin can be easily obtained from pine resins and some other plants, which has been used as a cheap feedstock chemical to replace petrochemicals in industry[33]. It is mainly composed of two types of rosin acids: abietic and pimaric acid. Among them, dehydroabietic acid (DA) belongs to the type of abietic acid, which is the largest and most stable component[34,35]. To increase the additional value of DA-based chemical products, studies on DA derivatives and their applications have been extensively carried out, of which the pharmaceutical research is under specific focus[36–41]. However, their application in optical functional materials have been scarcely investigated. From the structural point of view, the aniline part of DAMBA (a derivative of DA; vide infra) could be easily coupled with salicylaldehyde to yield Schiff bases, which usually owns flexible structure with photoisomerization property[42–45]. While the alicyclic (decalin) structure of DAMBA actually could be thought as a structural analog to phytosterol and cholesterol, which has been commonly used by plant and zooblast cells to stiffen the unsaturated lipid membrane and sustain the normal physiological functions[46]. Hence, structural marriage of the flexible Schiff bases with the rigid decalin structure of DAMBA potentially provides a strategy to regulate the excited-state molecular motion of Schiff bases and endows the generated molecules with unexpected photophysical properties or functions. Noteworthy, the luminescent properties correlate closely with their excited-state molecular motions, which unfortunately occurs in an ultrafast timescale that is too fast to trace and regulate. The fully understanding and regulation of the molecular motions on the excited state is thus of great significance for the rational and accurate design of new luminescent materials[47–51].

Herein, we have succeeded a series of alicycle-fused Schiff bases as a new class of BioAIEgens derived from natural rosin. The alicyclic moieties help to rigidify the flexible structure of the non-emissive Schiff base to light up its fluorescence. Furthermore, the luminescence properties could be facilely tuned by structural substitution since the substitution variations could easily affect their intermolecular interactions and their excited-state molecular motion. Such molecular motion distinction-resulted luminescence variation could also be monitored and visualized in the solid state by either fluorescence or photochromic properties. Finally, due to the lipid characteristic and biocompatibilities of these BioAIEgens, they were applied in cell imaging, exhibiting superior organelle selectivity in lipid droplet and lysosome, respectively. This work not only demonstrates a new renewable and sustainable source to acquire BioAIEgens and unveils the underlying structure–property relationship, but also provides a new rigidification strategy to improve the efficiency of luminogens.

## Results

**Molecular synthesis and characterization**. As shown in Fig. 1, DA extracted from disproportionated rosin could be chemically modified to its derivative DAMBA[41], which is herein applied as a precursor to react with a series of various substituted salicylaldehyde to yield their corresponding Schiff bases, DAMB-SA, DAMB-SAB, and DAMB-SAN, with a high crystalline yield of around 70% without chromatographic purification. The structures and purity of the obtained products were confirmed by melting point, NMR as well as high resolution mass spectroscopy measurements (Figs. S2–S11). Single crystals of all Schiff bases suitable for X-ray diffraction measurement could be easily obtained via slow evaporation in their respective solutions (please see the details in the "Methods" section).

**Aggregation-induced emission properties**. Figure 2a gives us an intuitive impression about the fluorescence properties of these alicycle-fused Schiff bases (Figs. 2b–d and S12). DAMB-SA and DAMB-SAB show no emission in acetonitrile (ACN) solutions but enhanced yellow and dull orange emissions when adding water into the ACN solutions, suggesting their typical AIE characteristic. As a comparison, DAMB-SAN is emissive in both the pure ACN solution and aggregates ($f_w = 90\%$), with green and greenish yellow light, respectively. DAMB-SAN thus shows aggregation-enhanced emission property. Dynamic light scattering (DLS) results (Fig. S13) support that nano aggregates of all

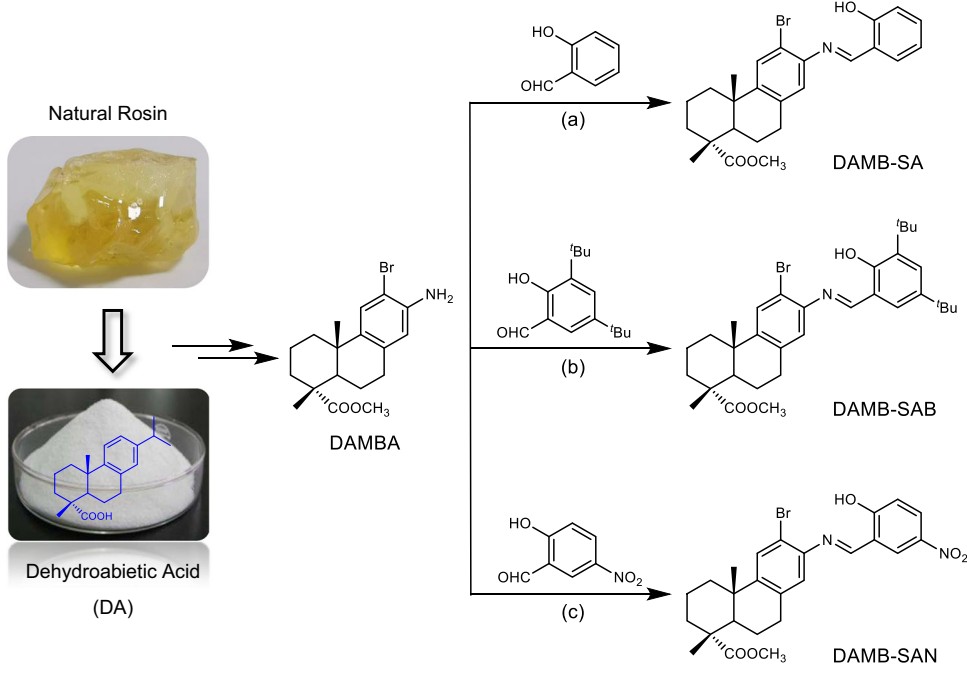

**Fig. 1 Schematic illustration of molecular synthesis.** Synthesis of dehydroabietic acid-derived Schiff bases: **a** EtOH, 90 °C, 10 h; **b** EtOH, 100 °C, over night; **c** EtOH, 85 °C, 3 h.

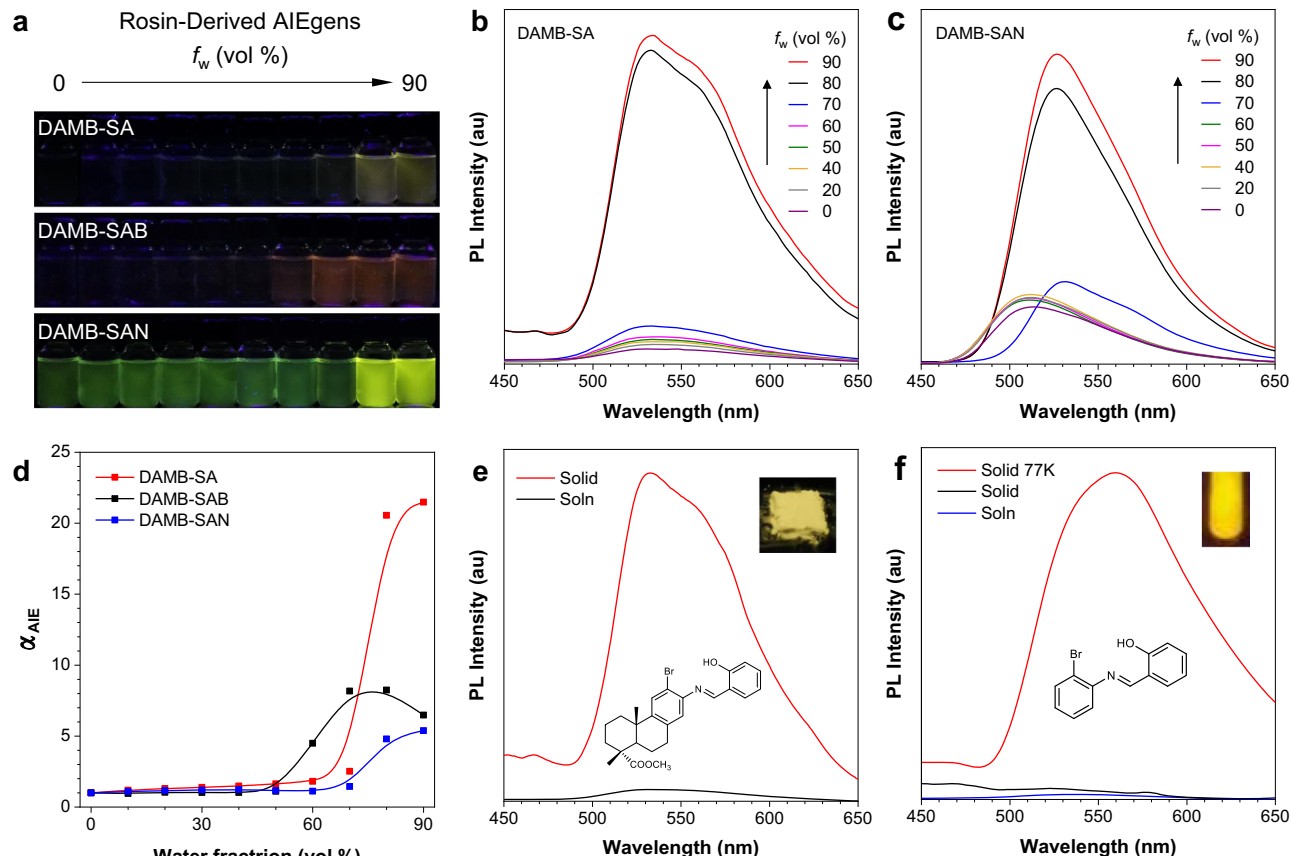

**Fig. 2 Aggregation-induced emission properties. a** Fluorescence photographs of DAMB-SA, DAMB-SAB, and DAMB-SAN in ACN/H₂O mixtures with different water fractions ($f_w$) taken under 365 nm UV irradiation. Concentration: 20 μM. **b, c** PL spectra of DAMB-SA (**b**) and DAMB-SAN (**c**) in ACN/H₂O mixtures with different $f_w$. Concentration: 20 μM; $\lambda_{ex}$: 350 nm. **d** The plots of the $\alpha_{AIE}$ versus the composition of the aqueous mixture of DAMB-SA, DAMB-SAB, and DAMN-SAN. $\alpha_{AIE} = I/I_0$, $I_0$ = PL intensity in pure ACN. **e** PL spectra of DAMB-SA in dilute ACN solution (20 μM) and as solid at room temperature, respectively. **f** PL spectra of AB-SA in dilute ACN solution (20 μM) and as solid at room temperature and 77 K, respectively. $\lambda_{ex}$: 350 nm. Inset: fluorescence photographs of DAMB-SA (room temperature) and AB-SA (77 K) taken under 365 nm UV irradiation. Concentration: 20 μM.

DA-derived Schiff bases are formed at high $f_w$ (80%), and their enhanced emission in ACN/H$_2$O mixtures is derived from their aggregation in poor solvents. Furthermore, concentration experiment also supports the AIE characteristic of these alicycle-fused luminogens since much enhanced emission can be observed at a higher concentration (Fig. S14). The discernible fluorescence colors inform us that the substituent variations must have played a critical role. Indeed, DFT calculation indicates that the substitution groups cause different band gaps of DAMB-SAB (3.87 eV), DAMB-SA (3.96 eV), and DAMB-SAN (3.98 eV), which matches well with their luminescence color change trend (Fig. S15). Additionally, no solvent effect can be found in the solvatochromic tests of DAMB-SAN (Fig. S16), indicating that charge transfer might not be the main reason to the emission wavelength variations. Besides their fluorescence color, the luminescence efficiency of the DA-derived AIEgens also shows obvious difference with solid-state quantum yields (QYs determined by an integrating sphere) of 1.5%, 0.3%, and 4.9% for DAMB-SA, DAMB-SAB, and DAMB-SAN, respectively. According to the AIE mechanism of restriction of intramolecular motion (RIM), the luminescence efficiency relates closely with the excited-state molecular motion that active excited-state molecular motion normally promotes the non-radiative decay channel to decrease the luminescence efficiency. Since DAMB-SA, DAMB-SAB, and DAMB-SAN share the same fundamental chromophoric structure of AB-SA, it is rational to compare the luminescence properties of the alicycle-fused AIEgens with their respective control molecules of AB-SA, AB-SAB, and AB-SAN to deeply understand their luminescence behaviors. Interestingly, although these alicycle-fused AIEgens are all AIE active with enhanced emission in the solid state (taking DAMB-SA as an example, Fig. 2e), AB-SA is surprisingly non-luminescent either at molecular or aggregate level (Fig. 2f). In addition, although the solid state of AB-SA is non-fluorescent at room temperature, it emits very strongly at 77 K. These results suggest that AB-SA has active excited-state molecular motion even in the solid state, which quenches its emission. While only in the ultralow temperature of 77 K, the molecular motion could be suppressed to generate emission. Furthermore, AB-SAB and AB-SAN exhibit weaker emission efficiencies than DAMB-SAB and DAMB-SAN (Fig. S17; QYs for AB-SAB, AB-SAN, DAMB-SAB, and DAMB-SAN: 0.1%, 0.2%, 0.3%, and 4.9%). Based on the above observation and further comparing the luminescence of control molecules with the alicycle-fused AIEgens, it is rational to speculate that the alicyclic moiety in the DA skeleton may help to rigidify the molecular conformation to result in the AIE property of these alicycle-fused AIEgens. However, the conformational rigidification effect of the alicyclic moiety on varied substituted AB-SA structures are also different, resulting in their variation in luminescence efficiency.

**Single crystal structure and crystal packing analysis**. In order to shed more light on the cause of their fluorescence distinctions and confirm our speculation, all alicycle-fused BioAIEgens together with the basic structure of AB-SA have been crystallographically measured and analyzed (Fig. 3 and Table S1). Among all, AB-SA has the more twisted structure and larger dihedral angle (46.03°) than DAMB-SAB (31.58°), DAMB-SA (28.36 and 18.44°), and DAMB-SAN (19.08 and 18.18°). More twisted molecular structure is unfavorable to stabilize the excited-state molecular conformation through electron delocalization and resonance. As a comparison, DAMB-SAN possesses the flattest conformation that helps to stabilize the excited molecular conformation, resulting in its luminescence behavior in solution state. Noteworthy, the intermolecular interactions of DAMB-SA, DAMB-SAB, and

DAMB-SAN in the crystal state also show interesting changes relevant to AB-SA. Overall, AB-SA exhibits the least intermolecular interactions with a relatively loose molecular arrangement and weak intermolecular restriction effect. This kind of molecular arrangement benefits the excited-state molecular motion, and thus can well explain the non-luminescence behavior of AB-SA in both solution and crystal state. However, when the alicyclic unit is fused on the molecular backbone, the intermolecular interaction becomes stronger, which has increased the restrain effect of the crystal arrangement of the BioAIEgens. And the restrain effect exhibits a proportional effect to the luminescence efficiency of these BioAIEgens. For example, according to the crystal arrangements in Fig. 3, the intermolecular interactions of DAMB-SAB, DAMB-SA, and DAMB-SAN exhibit a gradual increase tendency, which is in good consistence with their luminescence efficiency change (QYs of DAMB-SAB: 0.3%, DAMB-SA: 1.5%, and DAMB-SAN: 4.9%). What's more, the melting point tendency (DAMB-SAN: 216–217 °C, DAMB-SA: 156–157 °C, DAMB-SAB: 104–105 °C, and AB-SA: 85–86 °C) matches well with their intermolecular interaction degree from DAMB-SAN, DAMB-SA, DAMB-SAB to AB-SA. Hence, it is rational to conclude that the alicyclic moiety together with the substituent groups endow AB-SA either a better conjugation at molecular state or enhanced intermolecular interactions at aggregate state, resulting in the more restrained excited-state molecular motion and the AIE behavior. In order to further verify the role of molecular motion in determining their fluorescence properties, we took DAMB-SA as an example and studied the fluorescence changes upon its viscosity and temperature variations. As shown in Fig. S18, when glycerol is added to the DAMB-SA solution in an EtOH/glycerol mixture to increase its viscosity, the emission becomes stronger due to the suppression of the molecular motion. Similar to the viscosity variation, the emission of the DAMB-SA solution in ACN at 77 K is strong while that at room temperature is totally non-fluorescent (Fig. S19).

**Visualization of photochromic procedures**. Simultaneous regulation and visualization of the solid-state molecular motion is a challenge although it is significant to understand the structural design of molecular machines or photo-responsive smart materials[47–51]. Herein, the active molecular motion of AB-SA and the conformational rigidification effect of alicyclic moiety enable the regulation of the solid-state molecular motion of the generated BioAIEgens. Superior to the monitoring of structural variation from enol to keto under UV light by infrared absorption (Fig. S20), the photochromic property of these BioAIEgens makes it possible to visualize the solid-state molecular motion through color changes. As shown in Fig. 4a, AB-SA exhibits the fastest color darkening at room temperature under UV lightening while this procedure is slowed down at 77 K (Fig. S21), vividly demonstrating its fast excited-state molecular motion at room temperature. This can be spectrally verified by the fast change in the absorption spectra of AB-SA and its inset (Fig. 4c) in UV lightening process. When it is fused on an alicyclic unit (DAMB-SA), its ambient solid-state molecular motion has been significantly retarded, which can be seen from the much slower color change, suggesting the obvious rigidification effect of the alicyclic moiety. When the bulky *t*-butyl is bis-substituted on the salicyl part (DAMB-SAB), the color change under UV illumination turns faster again, informing us that the bulky and flexible substituent might play an anti-rigidification role in the excited state, resulting in its faster molecular motion. Specifically, if a rigid nitro group substitutes, barely no color change can be observed in DAMB-SAN because of its scarce molecular motion characteristic. Such color change variations of all alicycle-fused BioAIEgens

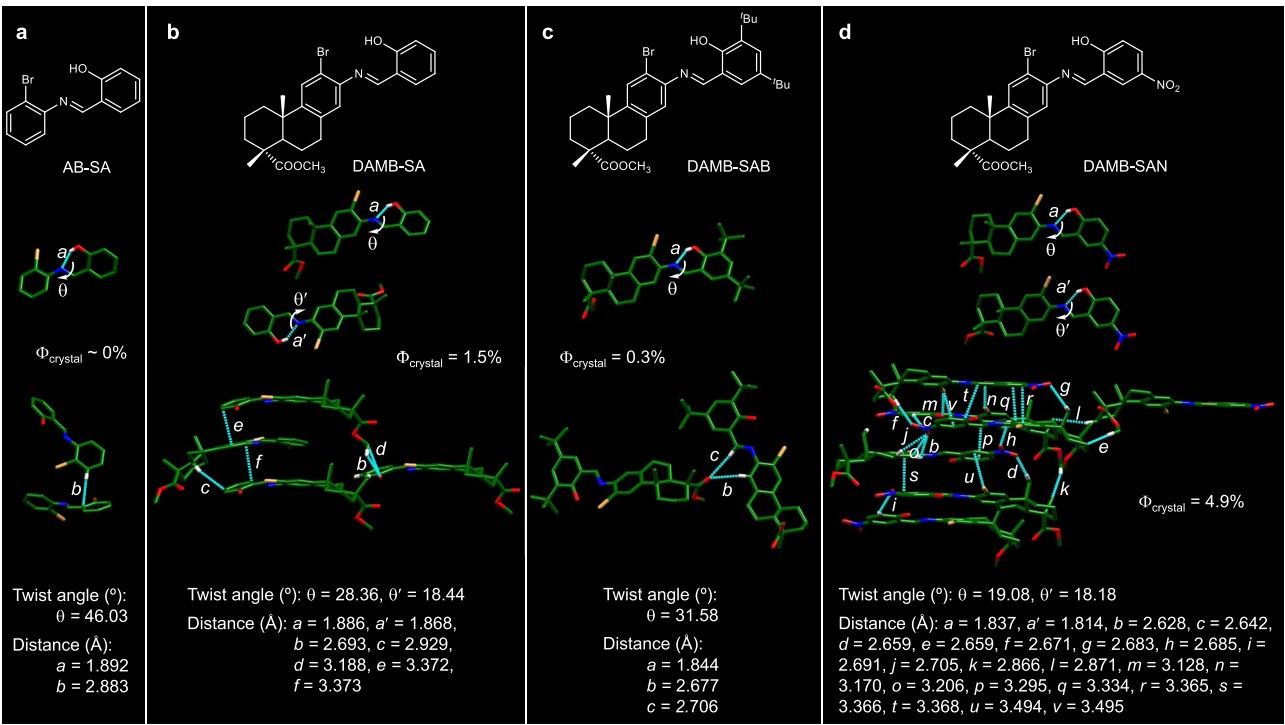

**Fig. 3 Single crystal structure and crystal packing.** Chemical structures (top), single crystal structures (middle), and their respective intermolecular interactions (bottom) of **a** AB-SA, **b** DAMB-SA, **c** DAMB-SAB, and **d** DAMB-SAN.

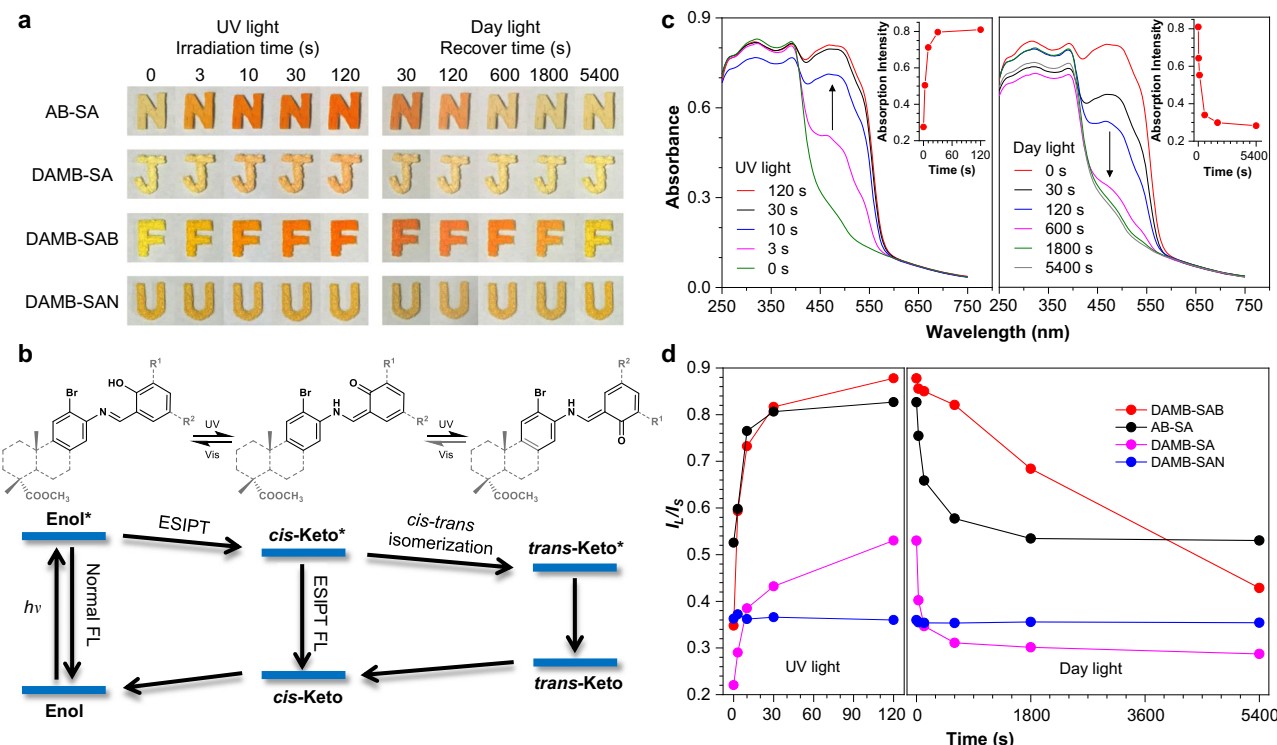

**Fig. 4 Visualization of photochromic procedures. a** Photochromic images and **b** the proposed reversible photochromic mechanism of AB-SA, DAMB-SA, DAMB-SAB, and DAMB-SAN upon UV and Day light irradiation. **c** The UV-DRS spectra of AB-SA in its reversible photochromic processes. Inset: The plots of the visible absorption maximum of AB-SA (469 nm) versus the irradiation and recover time, respectively. **d** The plots of the $I_L/I_S$ versus the irradiation and recover time, respectively. $I_L$ and $I_S$ represent the absorption intensity at the long and short wavelength in the visible region, respectively.

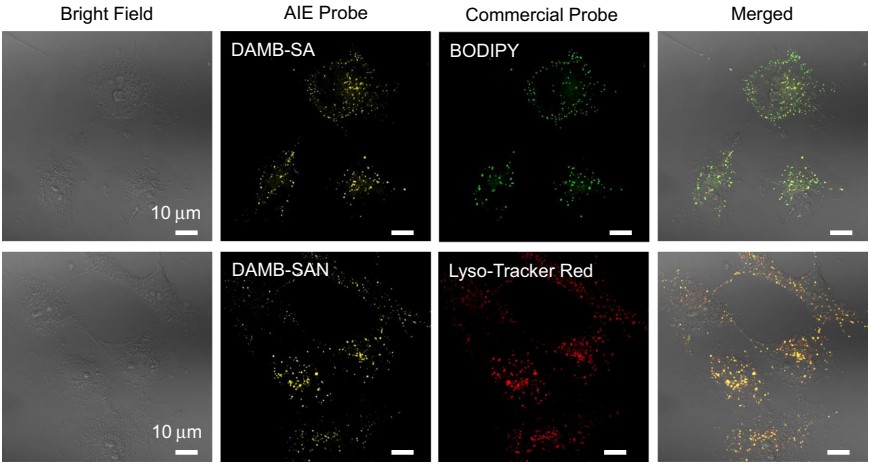

**Fig. 5 Targeted imaging.** CLSM images of COS-7 cells stained with DAMB-SA (10 μM), BODIPY (1 μM), DAMB-SAN (10 μM), and Lyso-Tracker Red (1 μM), respectively. DAMB-SA channel: $\lambda_{ex}$: 405 nm, $\lambda_{em}$: 540–600 nm; BODIPY channel: $\lambda_{ex}$: 488 nm, $\lambda_{em}$: 500–530 nm; DAMB-SAN channel: $\lambda_{ex}$: 405 nm, $\lambda_{em}$: 540–600 nm; Lyso-Tracker Red channel: $\lambda_{ex}$: 543 nm, $\lambda_{em}$: 580–650 nm.

can be spectrally found in Fig. S22, showing their color darkening sequence completely the same to their molecular motion behaviors. To gain a deeper understanding on the molecular motion behaviors, the fluorescence figures shoot at varied temperatures are shown in Fig. S23. Both AB-SA and DAMB-SAN show high contrast of the emission intensity at RT and 77 K, while the emission contrast of DAMB-SA and DAMB-SAN is smaller, indicating that both AB-SA and DAMB-SAB has a dynamic molecular motion at RT, resulting in the excited-state energy loss as non-radiative decay. Similarly, the solid-state X-ray diffraction patterns have confirmed this conclusion. As shown in Fig. S24, both AB-SA and DAMB-SAB exhibit nearly identical PXRD patterns as pristine and ground, while DAMB-SA and DAMB-SAN show distinct PXRD curves, indicating that during the time between grinding and immediate sample measuring, AB-SA and DAMB-SAB are already recrystallized[47]. As a result, AB-SA and DAMB-SAB have a faster molecular motion compared to DAMB-SA and DAMB-SAN. In addition, the decolorization phenomena in the recover process match well with their absorption spectra. Furthermore, the plots of $I_L/I_S$ ($I_L$ and $I_S$ are responsible for *trans*-Keto and *cis*-Keto absorption, respectively) versus the photochromic time (Fig. 4d), rightly depict the reversible photochromic procedures in Fig. 4a, demonstrating that *trans*-Keto overwhelms *cis*-Keto in cases of AB-SA and DAMB-SAB under UV light. The reason why DAMB-SAB has a slower turn-back procedure might be because the bulky *t*-butyl groups could restrain its *trans*-Keto to *cis*-Keto transformation at ambient condition. Based on their fundamental structure of AB-SA, it is reasonable to assume their reversible phtochromic procedure as shown in Fig. 4b, where an excited-state molecular motion occurs between the *cis*-Keto and *trans*-Keto forms[44,45]. Undoubtedly, the introduction of the alicyclic structure together with substituent variation could easily regulate their excited-state molecular motion behavior, that can be visualize by the photochromic rate from AB-SA, DAMB-SAB, DAMB-SA, to DAMB-SAN.

**Targeted imaging**. Since the alicyclic moiety originated from natural rosin may endow these BioAIEgens biocompatibility, we thus evaluated the cell and bacterial viability. As shown in Figs. S25 and S26, more than 90% viability in the COS-7 cell and *S. epidermidis* and *E. coli* could be remained even at high sample concentrations of 20 μM, suggesting the excellent biocompatibility of these BioAIEgens. Hence, we have explored the application of these BioAIEgens in COS-7 cell imaging, and the results

indicate that DAMB-SA can specifically stain lipid droplets (LDs), while DAMB-SAN prefers to stain lysosome (Fig. 5). The LDs selectivity of DAMB-SA is possibly ascribed to the lipotropy of its alicyclic moiety. On the other hand, the nitro group in DAMB-SAN is acidophilic[52], therefore preferring to enter the slightly acidic lysosome. In addition, the bacterial imaging (Fig. S26a, c) indicates that DAMB-SA and DAMB-SAN can stain both Grampositive (*S. epidermidis*) and Gram-negative (*E. coli*) bacteria. Such results have verified the universality of our BioAIEgens in bio-application. As a key factor of bioimaging, the photostability tests of these DAMB-SA and DAMB-SAN in COS-7 cells are also provided (Fig. S27). The fluorescence signals of the BioAIEgens slightly decrease to ~98% of their initial values, and the morphology of the cells is still clear after one-photon laser irradiation. In comparison, the fluorescence signals of BODIPY and Lyso-Tracker Red fade to ~73% and ~87% of their initial values during the same process, respectively. As a result, the BioAIEgens exhibit superior photostability, which is favorable for the bioimaging.

## Discussion

Natural product-derived compounds are one of the most significant bioactive nature-based compounds due to their natural availability, renewability, sustainability, biodegradability, and biocompatibility compared to the pure artificial chemicals[22]. Among them, rosin is a renewable and sustainable natural resource, which is nevertheless limited to low value-added applications in current industry. In order to add high values of the rosin-derived biomaterials, we have herein succeeded a new class of BioAIEgens from the abundant and low-cost natural rosin via facile Schiff base reactions. As expected, the introduction of the naturally alicyclic moiety favors the rigidification of the flexible AB-SA and suppression of excited-state molecular motion, hence endowing it with AIE characteristic. Furthermore, the substituent variation makes it possible to regulate their molecular motion behaviors. Such molecular motion discrepancies can be visualized by distinct photophysical properties: the ones with dynamic molecular motion are inclined to weaker solid-state emission and stronger photochromism, while those with restricted molecular motion prefers stronger solid-state emission and weaker photochromism. These BioAIEgens could respectively target lipid droplet and lysosome with high biocompatibility. Therefore, this work not only provides a new design hypothesis of AIEgens, but also endows the natural rosin with higher added-values.

## Methods

**Materials**. The natural product, dehydroabietic acid (DA), was obtained via reported purification procedures from the commercially disproportionated rosin (Guangxi Jinxiu Songyuan Forest Products Co., Ltd.)[53]. Its derivative which is synchronically used as the precursor in this work, DAMBA, was synthesized according to the procedures previously reported in the literature[41]. All kinds of substituted salicylaldehyde (Energy Chemical, 98%), 2-bromoaniline (Energy Chemical, 98%), and chloroform-$d_3$ (CDCl$_3$, J&K Scientific, 99.8%) were used without further purification. All the organic solvents were purchased from Nanjing Chemical Reagent Co., Ltd. and used without further purifications. Milli-Q water was from a Milli-Q purification system (Merck Millipore, Germany). Dulbecco's minimum essential medium (DMEM) and phosphate-buffered saline (PBS) were purchased from Gibco. Fetal bovine serum (FBS), penicillin, and streptomycin were purchased from Invitrogen. Luria-Bertani (LB) broth and LB agar were from USB Co. Zinc dust. *Escherichia coli* (*E. coli*) (ATCC 25922) and *Staphylococcus epidermidis* (*S. epidermidis*) (ATCC 12228) were from ATCC. COS-7 cells (ATCC® CRL-1651™) were from ATCC

**Instrumentation**. Melting points were determined on an OptiMelt MPA100 apparatus (SRS, USA) without correction. FT-IR spectra were recorded on a Thermo Scientific Nicolet Summit FT-IR Spectrometer (Thermo Scientific, USA) with KBr methods in the 4000–500 cm$^{-1}$ range. NMR measurements were performed on a Bruker AVANCE-III-600 spectrometer ($^1$H, 600 Hz; $^{13}$C, 150 Hz) with CDCl$_3$ as solvent unless otherwise stated. High resolution mass spectra (HRMS) were recorded on a GCT premier CAB048 mass spectrometer operating in a MALDI-TOF mode. Dynamic light scattering (DLS) measurements were performed on a Malvern Zetasizer Nano ZS analyzer. Powder X-ray diffraction (PXRD) experiment was performed on a Rigaku Ultima IV diffractometer with Cu Kα radiation. UV–Vis absorption spectra were recorded with a Shimadzu UV2450 spectrometer. UV–Vis diffuse reflectance spectra were recorded using a PE Lambda 950 UV–Vis–NIR spectrometer, wherein BaSO$_4$ was used as the reference. Photoluminescence (PL) spectra at room temperature were recorded on a Horiba Fluorolog-3 spectrofluorometer, while those at 77 K were obtained on an Edinburgh FLS980 fluorescence spectrophotometer equipped with a xenon arc lamp (Xe900). The absolute fluorescence quantum yields were determined on a Hamamatsu Quantaurus-QY C13534 spectrometer by a Quanta-φ integrating sphere. Molecular geometry optimization was calculated using the DFT method with the Gaussian 09 program package at the level of B3LYP/6-31G*. Single crystal data of DAMB-SA and DAMB-SAB were collected on a Bruker D8 VENTURE PHOTON 100 diffractometer using a graphite-monochromated Mo Kα radiation (0.71073 Å) in the ω–2θ scan mode. Nevertheless, single crystals of AB-SA and DAMB-SAN were selected and mounted on a SuperNova, Dual, Cu at zero, AtlasS2 diffractometer (Kα radiation: 1.54178 Å), which were kept at 293.1(2) and 100.00 (10) K, respectively, during data collection. Using Olex2[54]. the structure was solved with the ShelXT[55] structure solution program using intrinsic phasing and refined with the ShelXL[56] refinement package using least squares minimization.

**Cell culturing**. The COS-7 cells were cultured in DMEM (Gibco, USA) containing 10% FBS (Gibco, USA) and antibiotics (100 units mL$^{-1}$ penicillin and 100 μg mL$^{-1}$ streptomycin) in a 5% CO$_2$ humidity incubator at 37 °C. The cells were subcultured every two or three days.

**Cell imaging**. The COS-7 cells ($1 \times 10^6$ cells mL$^{-1}$) were seeded in confocal dish. After being cultured for 24 h, cells were added with 10 μM of DAMB-SA, DAMB-SAN in culture medium and incubated for 30 min at 37 °C. Then, cells were washed by PBS buffer (pH = 7.4) and imaged by using a confocal laser scanning microscope (CLSM) (LSM880, Carl Zeiss, Germany). Capture condition: $\lambda_{ex}$ = 405 nm, $\lambda_{em}$ = 540–600 nm.

The COS-7 cells were added with 10 μM of DAMB-SA in culture medium and incubated for 15 min at 37 °C, then added with 1 μM of BODIPY (Invitrogen, USA) and Lyso-Tracker Red (Invitrogen, USA) in culture medium and incubated for 15 min. Then, cells were washed by PBS buffer and imaged by CLSM. The COS-7 cells were added with 10 μM of DAMB-SAN in culture medium and incubated for 15 min at 37 °C, then added with 1 μM of Lyso-Tracker Red (Invitrogen, USA) in culture medium and incubated for 15 min. Then, cells were washed by PBS buffer and imaged by CLSM (LSM880, Carl Zeiss, Germany). Capture condition: BODIPY: $\lambda_{ex}$ = 488 nm, $\lambda_{em}$ = 500–530 nm; Lyso-Tracker Red: $\lambda_{ex}$: 543 nm, $\lambda_{em}$: 580–650 nm.

**Cytotoxicity assay**. The cell proliferation and cytotoxicity assay was carried out by 3-(4,5-dimethylthiazol-2-yl)-2,5-diphenyltetrazolium bromide (MTT) (Sigma, USA) as follows: COS-7 cells were cultured in 96-well plates ($8 \times 10^3$ cells per well) for 24 h, and then incubated with different concentrations (0, 2.5, 5, 10, and 20 μM) of DAMB-SA, DAMB-SAN for 24 h. The medium in each 96-well plate was removed completely and 100 μL of MTT mixed solution was added to each culture well, which consisted of 90 μL of culture medium and 10 μL of MTT solution (5 mg mL$^{-1}$). After incubation at 37 °C and in 5% CO$_2$ atmosphere for 4 h, 100 μL of DMSO was added to each well after the 100 μL MTT mixed solution removed. Finally, OD was measured using an enzyme mark instrument (Tecan, infinite F50,

Switzerland), at an optical absorbance of 570 nm. Five replicate measurements were obtained for each sample ($n = 5$). The survival rate of cells was determined by dividing the cell viability of the cells incubated with DAMB-SA, DAMB-SAN by the cell viability of the control group performed in the absence of DAMB-SA, DAMB-SAN.

**Bacterial culturing**. *E. coli* and *S. epidermidis* were cultured in the LB medium at 37 °C with a shaking speed of 200 revolutions per minute (rpm). Bacteria were harvested by centrifuging at 4722 relative centrifugal force (×g) for 3 min and washed twice. The optical density of the bacteria suspension was measured on a microplate reader (Tecan, infinite M200, Switzerland) at 600 nm.

**Bacterial staining**. Ten micromolar of DAMB-SA, DAMB-SAN solution in PBS containing $5 \times 10^8$ colony forming unit (CFU mL$^{-1}$) of *E. coli* and *S. epidermidis* was transferred to a 1.5 mL microcentrifuge tube. After dispersion via vortex, the bacteria were incubated at 37 °C with a shaking speed of 200 rpm for 30 min. To take fluorescence images, 1 μL of stained bacteria solution was transferred to a piece of glass slide and then covered by a coverslip. The images were collected using a CLSM (LSM710, Carl Zeiss, Germany). Capture condition: $\lambda_{ex}$ = 405 nm, $\lambda_{em}$ = 540–600 nm.

**Bacterial inhibition assay**. The MIC values of DAMB-SA and DAMB-SAN against *E. coli* and *S. epidermidis* were determined by broth microdilution according to the Clinical and Laboratory Standards Institute (CLSI) 2018 guideline[57]. The turbidity of bacteria upon treatments or not was recorded by the optical density value at 600 nm (OD600) using a microplate reader (Synergy™ H1, BioTek, USA). Bacteria viability (%) = (OD600$_{sample-24h}$ − OD600$_{sample-0h}$) / (OD600$_{control-24 h}$ − OD600$_{control-0h}$) × 100%[58]. The data were obtained from replicate experiments ($n = 3$).

**Photostability of DAMB-SA and DAMB-SAN**. Photostability of DAMB-SA (405 nm) and commercial dye BODIPY (488 nm), DAMB-SAN (405 nm) and commercial dye Lyso-Tracker Red (543 nm) in COS7 cells was measured under one-photon excitation with 1% laser power and statistically analyzed using image processing software (Origin 9.6.5.169 software). The data were obtained from replicate experiments ($n = 5$).

**Statistical analysis**. The values reported are expressed as mean standard deviation (SD). The Origin 9.6.5.169 and GraphPad Prism 8.0.2 softwares were used for graph plotting. Each experiment includes at least three replicates.

**Reporting summary**. Further information on research design is available in the Nature Research Reporting Summary linked to this article.

## Data availability

The authors declare that all the data supporting the findings of this manuscript are available within the manuscript and Supplementary Information files and also available from the corresponding authors upon reasonable request. The X-ray crystallographic coordinates for structures reported in this study have been deposited at the Cambridge Crystallographic Data Centre (CCDC), under deposition numbers of AB-SA (1974101), DAMB-SA (1963938), DAMB-SAB (1963939), and DAMB-SAN (2019547). These data can be obtained free of charge from The Cambridge Crystallographic Data Centre via www.ccdc.cam.ac.uk/data_request/cif. Source data are provided with this paper.

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

## Acknowledgements

Support from the National Natural Science Foundation of China (21601087 and 21805233) is gratefully acknowledged. We also thank the Science and Technology Plan of Shenzhen (JCYJ20180306174910791) and the Innovation and Technology Commission (ITC–CNERC14SC01 and ITCPD/17–9). The authors also acknowledge the Advanced Analysis & Testing Center of Nanjing Forestry University for testing services and Professor Tao Zeng for supplying us with his materials relevant to rosin-based research.

## Author contributions

X.-M.C., Z.Z., and B.Z.T. conceived and designed the experiments. Y.L. (Yuting Lin) and X.C. performed the synthesis. X.-M.C., Y.L. (Yuting Lin), and X.C. did the PL measurements and analyzed the data. Z.W. and X.Z. conducted QY measurements. Y.L. (Ying Li) performed the cell and bacterial culturing and imaging tests as well as cytotoxicity experiments. Z.Z. performed the theoretical calculation. X.-M.C., S.H., Z.Z., and B.Z.T. took part in the discussion and give important suggestions. X.-M.C., and Z.Z., co-wrote the paper.

## Competing interests

The authors declare no competing interests.
