## [Peer Review File · Nature Communications]

REVIEWER COMMENTS

Reviewer #1 (Remarks to the Author):

natural product-based AIEgens is extremely important and should have a wide application in biosystems because of their perfect bio-compatibility. In this research, the authors developed a new class of natural rosin-derived luminogens with aggregation-induced emission (AIE) property. The authors found that the as-prepared materials displayed good biocompatibility and targeted organelle imaging capability. In addition, the as-prepared natural product-based AIE luminogens (AIEgens) has been demonstrated to display photoirradiation-responsive luminescence change and photochromic behavior in the solid state. The research is very important, the paper is well-written and well-organized, I would like to recommend its publication after minor revisions.

(1) Please indicate if the solid-state quantum yields are absolute yield or related to one reference.

(2) Please discuss if the Br atom is necessary or not? How does this atom affect the AIE performance?

(3) In bio-application, what is the particle size? Do the particles affect the emission? If NOT, why? If yes, how

(4) Reference 14, the title of article

(5) Please double check the twisted angle. It might be not correct. First, all molecules might be easy to form N...H-O bonds (six-member ring), which is much stable. The twisted angle should be the angle between DAMB plane and SA(or SAB/SAN) unit.

Reviewer #2 (Remarks to the Author):

Exploration of AIE materials have attracted great attention from researchers. In this work, Tang and co-workers focus on addressing several fundamental issues in AIE research, such as the biocompatibility and eco-compatibility of the AIE materials besides the photophysical properties. The authors have successfully demonstrated the AIE properties of the natural rosin derivatives, which have shown photoirradiation-responsive luminescence change and photochromic behavior in solid states. Although the quantum yields of those materials are not high, the viewpoints of this work are quite new and inspiring as a fundamental research. Additionally, they used an alicyclic moiety to rigidify the molecular conformation to enhance the photoluminescence, which has been rarely reported. In general, the manuscript was well prepared and the conclusions can be supported by experimental data. This work should be suitable for publication in Nature Communications, after addressing the following concerns from the reviewer:

1. The authors used a new term of "bioAIEgen" in the title. The reviewer understands that this "artificial word" may be an important selling point of this work. However, the authors should further elaborate the definition and advantages of such BioAIEgen in the manuscript, which is very important to help the readers understand the superiority of such natural product-derived AIEgens.

2. The authors claimed that the obtained AIEgens are biocompatible and eco-compatible. However, the reviewer only noticed the results of biocompatibility study in the manuscript. The authors should also test the degradation capability of these AIEgens under varied conditions to demonstrate their

eco-compatibility.

3. The authors have argued that molecular motion in the solid state is responsible for the non-emissive property of AB-SA and weak emission of DAMB-SAB. Besides the photochromic change, more data should be provided to support this conclusion.
4. The three compounds have exhibited different emission colors, suggesting that their electronic structures may be influenced by the substitution. To better understand the variation in their emission wavelengths, the solvatochromic effect should be investigated to confirm whether the charge transfer has any influence on their emission profiles.
5. As far as the reviewer knows, dehydroabiatic acid (DA) only presents a small percentage in the natural rosin. In the Molecular synthesis and characterization part, the authors claimed that "As shown in Fig. 1, DA extracted from natural rosin could be chemically modified to its derivative DAMBA". In the Experimental section, they described that the DA was obtained via reported purification procedures from the commercially disproportionated rosin. The authors should rephrase the description in manuscript to avoid any misunderstanding.
6. The results indicate that the introduction of alicyclic moiety can rigidify the flexibility of AB-SA, resulting in the AIE property of DAMB-SA. However, the control molecules of DAMB-SAB and DAMB-SAN are not provided. The author is suggested to provide the data for control molecules of DAMB-SAB and DAMB-SAN, which can better demonstrate the rigidification effect of alicyclic moiety.
7. The authors claimed that the good biocompatibility is a significant advantage of these natural rosin based AIEgens. However, only the cell imaging application has been implemented. Are there other biological applications that can support their claims, for example, any in vivo evaluation?
8. As a key factor for bioimaging, the photostability of these AIEgens should also be provided.
9. Some minor changes should be made in the manuscript. For example, "excited-state molecular motion" should be "excited-state molecular motion" in page 4; "alicyclic moiety originated from natural rosin" should be "alicyclic moiety originated from natural rosin" in page 5. The authors claimed that "these luminogens generally face problems....., poor biocompatibility, and awful biodegradability". The authors are suggested to rephrase this statement, since a number of synthetic luminogens have shown acceptable or even good biocompatibility in general biological applications.
10. In the discussion part, the authors are encouraged to elaborate more about the superiorities of natural product-derived luminogens in comparison to the pure artificial/synthetic ones.
11. Please indicate which statistical analysis method was employed in the biological analysis.

Reviewers' comments for the manuscript titled **“BioAIEgens Derived from Rosin: How Does Molecular Motion Affect Their Photophysical Processes in Solid State?”**

Reviewer #1 (Remarks to the Author):

Natural product-based AIEgens is extremely important and should have a wide application in biosystems because of their perfect biocompatibility. In this research, the authors developed a new class of natural rosin-derived luminogens with aggregation-induced emission (AIE) property. The authors found that the as-prepared materials displayed good biocompatibility and targeted organelle imaging capability. In addition, the as-prepared natural product-based AIE luminogens (AIEgens) has been demonstrated to display photoirradiation-responsive luminescence change and photochromic behavior in the solid state. The research is very important, the paper is well-written and well-organized, I would like to recommend its publication after minor revisions.

1. Please indicate if the solid-state quantum yields are absolute yield or related to one reference.
2. Please discuss if the Br atom is necessary or not? How does this atom affect the AIE performance?
3. In bio-application, what is the particle size? Do the particles affect the emission? If not, why? If yes, how?
4. Reference 14, the title of article.
5. Please double check the twisted angle. It might be not correct. First, all molecules might be easy to form $N\cdots H-O$ bonds (six-member ring), which is much stable. The twisted angle should be the angle between DAMB plane and SA (or SAB/SAN) unit.

Reviewer #2 (Remarks to the Author):

Exploration of AIE materials have attracted great attention from researchers. In this work, Tang and co-workers focus on addressing several fundamental issues in AIE research, such as the biocompatibility and eco-compatibility of the AIE materials besides the photophysical properties. The authors have successfully demonstrated the AIE properties of the natural rosin derivatives, which have shown photoirradiation-responsive luminescence change and photochromic behavior in solid states. Although the quantum yields of those materials are not high, the viewpoints of this work are quite new and inspiring as a fundamental research. Additionally, they used an alicyclic moiety to rigidify the molecular conformation to enhance the photoluminescence, which

has been rarely reported. In general, the manuscript was well prepared and the conclusions can be supported by experimental data. This work should be suitable for publication in Nature Communications, after addressing the following concerns from the reviewer:

1. The authors used a new term of “BioAIEgen” in the title. The reviewer understands that this “artificial word” may be an important selling point of this work. However, the authors should further elaborate the definition and advantages of such BioAIEgen in the manuscript, which is very important to help the readers understand the superiority of such natural product-derived AIEgens.
2. The authors claimed that the obtained AIEgens are biocompatible and eco-compatible. However, the reviewer only noticed the results of biocompatibility study in the manuscript. The authors should also test the degradation capability of these AIEgens under varied conditions to demonstrate their eco-compatibility.
3. The authors have argued that molecular motion in the solid state is responsible for the non-emissive property of AB-SA and weak emission of DAMB-SAB. Besides the photochromic change, more data should be provided to support this conclusion.
4. The three compounds have exhibited different emission colors, suggesting that their electronic structures may be influenced by the substitution. To better understand the variation in their emission wavelengths, the solvatochromic effect should be investigated to confirm whether the charge transfer has any influence on their emission profiles.
5. As far as the reviewer knows, dehydroabietic acid (DA) only presents a small percentage in the natural rosin. In the Molecular synthesis and characterization part, the authors claimed that “As shown in Fig. 1, DA extracted from natural rosin could be chemically modified to its derivative DAMBA”. In the Experimental section, they described that the DA was obtained via reported purification procedures from the commercially disproportionated rosin. The authors should rephrase the description in manuscript to avoid any misunderstanding.
6. The results indicate that the introduction of alicyclic moiety can rigidify the flexibility of AB-SA, resulting in the AIE property of DAMB-SA. However, the control molecules of DAMB-SAB and DAMB-SAN are not provided. The author is suggested to provide the data for control molecules of DAMB-SAB and DAMB-SAN, which can better demonstrate the rigidification effect of alicyclic moiety.
7. The authors claimed that the good biocompatibility is a significant advantage of these natural rosin based AIEgens. However, only the cell imaging application has been implemented. Are

there other biological applications that can support their claims, for example, any in vivo evaluation?

8. As a key factor for bioimaging, the photostability of these AIEgens should also be provided.

9. Some minor changes should be made in the manuscript. For example, “excited-state molecular motion” should be “excited-state molecular motion” in page 4; “alicyclic moiety originated from natural rosin” should be “alicyclic moiety originated from natural rosin” in page 5. The authors claimed that “these luminogens generally face problems....., poor biocompatibility, and awful biodegradability”. The authors are suggested to rephrase this statement, since a number of synthetic luminogens have shown acceptable or even good biocompatibility in general biological applications.

10. In the discussion part, the authors are encouraged to elaborate more about the superiorities of natural product-derived luminogens in comparison to the pure artificial/synthetic ones.

11. Please indicate which statistical analysis method was employed in the biological analysis.

Responses to reviewers' comments for the manuscript titled **“BioAIEgens Derived from Rosin: How Does Molecular Motion Affect Their Photophysical Processes in Solid State?”**

Reviewer 1:

Comments:

Natural product-based AIEgens is extremely important and should have a wide application in biosystems because of their perfect biocompatibility. In this research, the authors developed a new class of natural rosin-derived luminogens with aggregation-induced emission (AIE) property. The authors found that the as-prepared materials displayed good biocompatibility and targeted organelle imaging capability. In addition, the as-prepared natural product-based AIE luminogens (AIEgens) has been demonstrated to display photoirradiation-responsive luminescence change and photochromic behavior in the solid state. The research is very important, the paper is well-written and well-organized, I would like to recommend its publication after minor revisions.

Responses: We sincerely thank the appreciation and recognition of the reviewer on our work. We have provided a detailed point-to-point response to each question.

1. Please indicate if the solid-state quantum yields are absolute yield or related to one reference.

Responses: We've measured the absolute solid-state quantum yields on a Hamamatsu Quantaurus-QY C13534 spectrometer by a Quanta- ϕ integrating sphere, which has been described in the Experimental Section of the Supporting Information. In order to make it more easy to capture the characterization condition of fluorescence efficiency, we've revised the text in the section of "Aggregation-induced emission properties" as "Besides their fluorescence color, the luminescence efficiency of the DA-derived AIEgens also shows obvious difference with solid-state quantum yields (QYs determined by an integrating sphere) of 1.5%, 0.3% and 4.9% for DAMB-SA, DAMB-SAB, and DAMB-SAN, respectively."

2. Please discuss if the Br atom is necessary or not? How does this atom affect the AIE performance?

Responses: We understand the concern of the reviewer that the Br atom might exhibit heavy-atom effect on the fluorescence property of the resultant molecules. Since the precursor of DAMBA with Br atom applied in our work is a previously reported molecule, we would like to

develop some natural rosin based optical biomaterials based on DAMBA. The Br atom of DAMBA originates from the chemical modification of DA to approach high value chemicals or drugs. In our project, we just use DAMBA as our starting material, the Br atom thus is not introduced by design. Since the reviewer is interested in the influence of Br atom on the AIE property of these BioAIEgens, we thus do some primary investigation to try to explore the role of Br in affecting the photoluminescence of the Schiff base. Because the control molecule of DAMB-SA without Br is difficult to synthesize, we thus took AB-SA as model compound to synthesize its control molecule as A-SA without Br atom substitution (Figure 1). Unfortunately, both AB-SA and A-SA are non-emissive at solid state, although the heavy atom effect of Br is generally thought unfavorable for fluorescence. We think more investigations are necessary to elucidate the effect of Br atom completely, which possibly will be implemented in our future research.

Compound	AB-SA	A-SA
Structure		Day light		UV light		QY (%)	~0	~0

Figure 1. Figures under Day light and UV light and QYs of AB-SA and A-SA.

3. In bio-application, what is the particle size? Do the particles affect the emission? If not, why? If yes, how?

Responses: As suggested, we've measured the particle sizes of DAMB-SA (10 μ M) and DAMB-SAN (10 μ M) in the cell-based DEME-FBS (DF) and bacterial-based LB mediums, respectively (Figure 2). DAMB-SA-DF and DAMB-SAN-DF have an average particle sizes of approximate 129 and 104 nm, while DAMB-SA-LB and DAMB-SAN-LB have an average particle sizes of around 143 and 178 nm, respectively. In addition, we have measured the particle sizes of all BioAIEgens (Figure 3) in ACN/H₂O mixtures with f_w from 60% to 90% to give some

informational conclusions. Particle sizes of around 200 nm for each BioAIEgen can be always obtained in each ACN/H₂O mixture, which has the similar size to those obtained in the bio-systems, indicating the formation of aggregates in the emissive mixtures. With regard to the relationship between the α_{AIE} and particle sizes (Figure 3), no clear variation trend can be found. We speculate that the emission performance of the nano-sized samples in ACN/H₂O mixtures may depend on the crystallinity of the particles besides their aggregate size. Some previously reported results also support this speculation. (*Chem. Commun.*, **2007**, 3255; *Nanoscale*, **2016**, 8, 15173-15180; and *Chem. Sci.*, **2017**, 8, 5440-5446.).

Figure 2. (A) Particle sizes and (B) the fluorescence intensity of DAMB-SA (10 μM) and DAMB-SAN (10 μM) in DEME-FBS and LB mediums, respectively.

Figure 3. The plots of the α_{AIE} and average particle sizes formed in ACN/H₂O mixtures (20 μM) by DLS versus the composition of the aqueous mixture of DAMB-SA (A), DAMB-SAB (B), and DAMB-SAN (C). $\alpha_{\text{AIE}} = I/I_0$, I_0 = PL intensity in pure ACN.

4. Reference 14, the title of article.

Responses: Thanks for the reviewer’s careful examination. We will revise it as following: “14. Weiss, J. FLUORESCENCE OF ORGANIC MOLECULES*. *Nature* **152**, 176-178 (1943).” will be revised to “14. Weiss, J. Fluorescence of Organic Molecules*. *Nature* **152**, 176-178 (1943).”

5. Please double check the twisted angle. It might be not correct. First, all molecules might be easy to form N···H-O bonds (six-member ring), which is much stable. The twisted angle should be the angle between DAMB plane and SA (or SAB/SAN) unit.

Responses: We understand the reviewer’s concern about the accuracy of the structural definition of twist angle that is always variedly defined by different research groups. We used the torsion angle of $C_{SA}-N-C-C_{non\ Br-bonded}$ in our case to define the twist angle, which can be referred to the publications of *J. Mater. Chem. B*, **2017**, *5*, 1650-1657 and *Mater. Chem. Front.*, **2021**, doi: 10.1039/D0QM00754D. In order to understand the twist angle distinctions among variable definitions, we have additionally measured the twist angles via definition of dihedral angles (Table 1). Taking AB-SA as an example (Figure 4), the twist angle defined as the dihedral angle between the brominated phenyl ring (Ph_{Br}) and the salicyl phenyl ring (Ph_{SA}) by the reviewer is only a few larger than those measured by the torsion angles in AB-SA, DAMB-SA, and DAMB-SAB, but almost the same in DAMB-SAN, demonstrating that twist angles measured either by torsion or dihedral angles are of rare distinctions. In order to be consistent with the measuring methods applied in many of the published papers in our group, we prefer to use the torsion angle as the twist angle in our current work.

Table 1. Twist angles found via variable definitions.

			AB-SA	DAMB-SA	DAMB-SAB	DAMB-SAN
Twist angle (°)	Torsion angle (°)	$C_{SA}-N-C-C_{non\ Br-bonded}$	46.03	28.36, 18.44	31.58	19.08, 18.18
	Dihedral angle (°)	Ph_{Br} vs Ph_{SA}	49.63	23.94, 21.37	33.30	19.08, 18.11

Figure 4. Twist angles measured via torsion and dihedral angle exemplified in AB-SA.

Reviewer 2:

Comments:

Exploration of AIE materials have attracted great attention from researchers. In this work, Tang and co-workers focus on addressing several fundamental issues in AIE research, such as the biocompatibility and eco-compatibility of the AIE materials besides the photophysical properties. The authors have successfully demonstrated the AIE properties of the natural rosin derivatives, which have shown photoirradiation-responsive luminescence change and photochromic behavior in solid states. Although the quantum yields of those materials are not high, the viewpoints of this work are quite new and inspiring as a fundamental research. Additionally, they used an alicyclic moiety to rigidify the molecular conformation to enhance the photoluminescence, which has been rarely reported. In general, the manuscript was well prepared and the conclusions can be supported by experimental data. This work should be suitable for publication in Nature Communications, after addressing the following concerns from the reviewer:

Responses: We are grateful for the detailed suggestions and valuable comments of the reviewer and appreciate very much the thorough examination of our manuscript. We have provided a detailed point-to-point response to each question.

1. The authors used a new term of “BioAIEgen” in the title. The reviewer understands that this “artificial word” may be an important selling point of this work. However, the authors should further elaborate the definition and advantages of such BioAIEgen in the manuscript, which is very important to help the readers understand the superiority of such natural product-derived AIEgens.

Responses: Thanks to the reviewer's suggestion and we've clearly described the definition of "BioAIEgen" and elaborated the advantages of such BioAIEgen in the introduction part as "Alternatively, plant-originated chemicals are green and bioactive compounds that have been widely studied as one of the most interesting nature-based compounds in the advanced (bio)sensing fields because of their numerous inherent and unique characteristics, such as natural availability, renewability, sustainability, biodegradability, and biocompatibility.²² Hence, AIEgens from natural resources have recently gained researchers' intense attention (Figure S1).²³⁻³² To date, some natural product-based AIEgens including natural polymers (sodium alginate and bovine serum albumin) and small molecules (quercetin, myricetin, and berberine chloride) have been investigated with unique advantages in bioapplications. Due to their advantageous biocompatibility, imaging can be applied either in vitro or in vivo. Nevertheless, in general, the development of these bioproduct-inspired AIEgens (BioAIEgens: AIEgens or AIE-active nano agents obtained from natural resources including natural products or derivatives by modifying natural products)²¹ is still in its infant and the limited variety of AIEgens from natural resources hinders their further investigation and development. In consideration of these, BioAIEgens derived from the chemical modification on the abundant and low-cost natural resources might be a good alternate to either pure natural or artificial AIEgens." Please see the related data and rephrased text in the revised manuscript.

2. The authors claimed that the obtained AIEgens are biocompatible and eco-compatible. However, the reviewer only noticed the results of biocompatibility study in the manuscript. The authors should also test the degradation capability of these AIEgens under varied conditions to demonstrate their eco-compatibility.

Responses: We thank the reviewer for the insightful suggestion. The key points of this work is to highlight the advantages of biocompatibility, renewability and sustainability of the natural resource derived AIEgens compared to pure artificial ones. We are also interested in the biodegradability of the obtained AIEgens and the degradation tests are therefore carried out as suggested. As shown on the TLC plates (Figure 5), all BioAIEgen solutions in acetonitrile turned degraded in acidic conditions, demonstrating their sensitivity to the acids. In addition, DAMB-SAN was selected to test its structural stability in acetonitrile solutions with varied acidic concentration due to its emissive property in acetonitrile solutions. Figure 5B exhibits the

degradation results of DAMB-SAN immediately and in one hour, demonstrating that DAMB-SAN can be degraded with the gradual increase of acid and time.

Figure 5. Degradation of all BioAIEgens under acidic conditions: (A) TLC figures of DAMB-SA, DAMB-SAB, and DAMB-SAN on acidic silica gel plate. (B) Figures of DAMB-SAN under Day light and UV light in acetonitrile solutions with varied acidic concentration at varied time (0 min and 1 h).

3. The authors have argued that molecular motion in the solid state is responsible for the non-emissive property of AB-SA and weak emission of DAMB-SAB. Besides the photochromic change, more data should be provided to support this conclusion.

Responses: Besides the photochromic change, their fluorescence figures at RT and 77 K as well as their PXRD results are shown in Figures 6 and 7. As we can see in Figure 6, both AB-SA and DAMB-SAN show high contrast of the emission intensity at RT and 77 K, while the emission contrast of DAMB-SA and DAMB-SAN is smaller, indicating that both AB-SA and DAMB-SAB has a dynamic molecular motion at RT, resulting in the excited-state energy loss as non-radiative decay. Similarly, the solid-state X-ray diffraction patterns have confirmed this

conclusion. As shown in Figure 7, both AB-SA and DAMB-SAB exhibit nearly identical PXRD patterns as pristine and ground, while DAMB-SA and DAMB-SAN show distinct PXRD curves, indicating that during the time between grinding and immediate sample measuring, AB-SA and DAMB-SAB are already recrystallized, which has been similarly reported in *Angew. Chem. Int. Ed.*, **2019**, *58*, 4536-4540. As a result, the supplementary data herein have undoubtedly confirmed the conclusion that molecular motion in the solid state is responsible for the non-emissive property of AB-SA and weak emission of DAMB-SAB. The related data and discussions have been involved in the revised manuscript and Supporting Information.

Figure 6. Fluorescence figures of (A) AB-SA, (B) DAMB-SAB, (C) DAMB-SA, and (D) DAMB-SAN at RT and 77 K.

Figure 7. PXRD patterns of (A) AB-SA, (B) DAMB-SAB, (C) DAMB-SA, and (D) DAMB-SAN as pristine and ground.

4. The three compounds have exhibited different emission colors, suggesting that their electronic structures may be influenced by the substitution. To better understand the variation in their emission wavelengths, the solvatochromic effect should be investigated to confirm whether the charge transfer has any influence on their emission profiles.

Responses: We agree that the variable substitution might influence on the electronic structures of the corresponding compounds, hence exhibiting variable emission colors. According to the

reviewer's suggestion, we have carried out the solvent effect tests of DAMB-SAN (Figure 8). No solvent effect can be found in the solvatochromic tests, indicating that charge transfer might not be the reason to the emission wavelength variations. According to our previous work (*Angew. Chem. Int. Ed.*, 2020, 59, 9888-9907), the crystalline packing tightness can impose significant effect on the emission performance. When the molecules are loosely packed (DAMB-SAB), the excited-state minimum with a small band gap is accessible energetically because of stronger molecular motion, resulting more in the non-radiative decay, thus leading to the weaker emission in the red-shifted region. With regard to the tightly packed molecules (DAMB-SA and DAMB-SAN), their excited-state minimum is not energetically accessible, resulting in the reduction of non-radiative decay, thus leading to the stronger emission in the blue-shifted region. In this work, the emission wavelengths of the solid-state BioAIEgens are 528 (DAMB-SAN), 532 (DAMB-SA), and 568 nm (DAMB-SAB), respectively, and their emission intensity decreases from DAMB-SAN, DAMB-SA, to DAMB-SAB, demonstrating that the fluorescence wavelength and intensity values are in good consistence with their packing modes. In summary, our work has confirmed that the packing effect has played the most significant role in regulating their emission wavelength and intensity. The related data and discussions have been involved in the revised manuscript and Supporting Information.

Figure 8. (A) Absorption and (B) PL spectra of DAMB-SA, DAMB-SAB, and DAMB-SAN in solvents with different polarities. Concentration: 20 μM . The absorption maximum of each solution was chosen as its excitation wavelength.

5. As far as the reviewer knows, dehydroabietic acid (DA) only presents a small percentage in the natural rosin. In the Molecular synthesis and characterization part, the authors claimed that

“As shown in Fig. 1, DA extracted from natural rosin could be chemically modified to its derivative DAMBA”. In the Experimental section, they described that the DA was obtained via reported purification procedures from the commercially disproportionated rosin. The authors should rephrase the description in manuscript to avoid any misunderstanding.

Responses: Thanks for the reviewer’s careful examination and we have revised the description as suggested. The sentence of “As shown in Fig. 1, DA extracted from natural rosin could be chemically modified to its derivative DAMBA,⁴⁰” has been revised to “As shown in Fig. 1, DA extracted from disproportionated rosin could be chemically modified to its derivative DAMBA,⁴⁰”.

6. The results indicate that the introduction of alicyclic moiety can rigidify the flexibility of AB-SA, resulting in the AIE property of DAMB-SA. However, the control molecules of DAMB-SAB and DAMB-SAN are not provided. The author is suggested to provide the data for control molecules of DAMB-SAB and DAMB-SAN, which can better demonstrate the rigidification effect of alicyclic moiety.

Responses: We agree that the emission data for the control molecules of DAMB-SAB and DAMB-SAN are necessary to better demonstrate the rigidification effect of the alicyclic moiety. As shown in Figure 9, the brighter emission figures as well as larger QY values of DAMB-SAB and DAMB-SAN over their control molecules of AB-SAB and AB-SAN have again verified our demonstration that the alicyclic moiety has rigidified the flexible Schiff bases, especially in the cases of DAMB-SA&AB-SA and DAMB-SAN&AB-SAN. The related data and discussions have been involved in the revised manuscript and Supporting Information.

Figure 9. Figures and QYs of (A) DAMB-SA and AB-SA, (B) DAMB-SAB and AB-SAB, and (C) DAMB-SAN and AB-SAN under Day light and UV light, respectively.

7. The authors claimed that the good biocompatibility is a significant advantage of these natural rosin based AIEgens. However, only the cell imaging application has been implemented. Are

there other biological applications that can support their claims, for example, any in vivo evaluation?

Responses: We appreciate the insightful suggestion of the reviewer. Unfortunately, the short emission wavelengths of DAMB-SA and DAMB-SAN in this study are hard to be applied in vivo imaging since autofluorescence always produces substantial background noise (*Nat. Nanotechnol.* **2009**, *4*, 710-711). Hence, we have also carried out the bacterial-based biological tests, including the bacterial imaging and viability tests (Figure 10), indicating that DAMB-SA and DAMB-SAN can stain both of Gram-positive (*S. epidermidis*) and Gram-negative (*E. coli*) bacteria and the toxicity results of DAMB-SA and DAMB-SAN toward bacteria demonstrate their nearly negligible toxicity. Such results have verified the universality of our BioAIEgens in bio-application. The related data and discussions have been involved in the revised manuscript and Supporting Information.

Figure 10. (A and C) CLSM images of (A) *S. epidermidis* and (C) *E. coli* stained with 10 μM of DAMB-SA and DAMB-SAN for 30 min, respectively. λ_{ex} : 405 nm, λ_{em} : 540-600 nm. (B and D) Bacterial viabilities of (B) *S. epidermidis* and (D) *E. coli* in the presence of different concentrations of DAMB-SA (red) and DAMB-SAN (blue), respectively.

8. As a key factor for bioimaging, the photostability of these AIEgens should also be provided.

Responses: Thanks for the reviewer's suggestion. The photostability tests of DAMB-SA and DAMB-SAN in COS-7 cells are supplemented, which is then compared with photostability results of the commercial dyes. As shown in Figure 11, the fluorescence signals of the BioAIEgens slightly decrease to ~98% of their initial values, and the morphology of the cells is still clear after one-photon laser irradiation. In comparison, the fluorescence signals of BODIPY and Lyso-Tracker Red fade to ~73% and ~87% of their initial values during the same process, respectively. In conclusion, the high photostability of the BioAIEgens is favorable for in vitro bioimaging. The related data and discussions have been involved in the revised manuscript and Supporting Information.

Figure 11. Photostability of DAMB-SA (10 μ M, yellow) and BODIPY (1 μ M, green) (A), DAMB-SAN (10 μ M, yellow) and Lyso-Tracker Red (1 μ M, red) (B) in COS7 cells under continuous one-photon laser irradiation. DAMB-SA channel: λ_{ex} : 405 nm, λ_{em} : 540-600 nm; BODIPY channel: λ_{ex} : 488 nm, λ_{em} : 500-530 nm. DAMB-SAN channel: λ_{ex} : 405 nm, λ_{em} : 540-600 nm; Lyso-Tracker Red channel: λ_{ex} : 543 nm, λ_{em} : 580-650 nm. Inset: Fluorescence images of COS7 cells with increasing number of scans.

9. Some minor changes should be made in the manuscript. For example, “excitd-state molecular motion” should be “excited-state molecular motion” in page 4; “alicyclic moiety originated form natural rosin” should be “alicyclic moiety originated from natural rosin” in page 5. The authors claimed that “these luminogens generally face problems....., poor biocompatibility, and awful biodegradability”. The authors are suggested to rephrase this statement, since a number of synthetic luminogens have shown acceptable or even good biocompatibility in general biological applications.

Responses: Thanks for the reviewer's careful reviewing and we have revised the typewriting mistakes accordingly. "Excitd-state molecular motion" has be changed to "excited-state molecular motion" in page 4 and "alicyclic moiety originated form natural rosin" has been revised to "alicyclic moiety originated from natural rosin" in page 5. Besides, some other minor typewriting mistakes have been checked out and revised. Please see the revised text marked in yellow in the revised manuscript.

We agree that the phrasing we have claimed that "However, these luminogens generally face problems like weakened emission in the aggregated state, poor biocompatibility, and awful biodegradability" sounds not particularly appropriate. We have revised the phrase as "However, many of them face problems like weakened emission in the aggregated state as well as poor renewability and sustainability."

10. In the discussion part, the authors are encouraged to elaborate more about the superiorities of natural product-derived luminogens in comparison to the pure artificial/synthetic ones.

Responses: Thanks to the reviewer's suggestion and we've revised the Discussion by elaborating more about the superiorities of natural product-derived luminogens in comparison to the pure artificial/synthetic ones as "Natural product-derived compounds are one of the most significant bioactive nature-based compounds due to their natural availability, renewability, sustainability, biodegradability, and biocompatibility compared to the pure artificial chemicals.²² Among them, rosin is a renewable and sustainable natural resource, which is nevertheless limited to low value-added applications in current industry. In order to add high values of the rosin-derived biomaterials, we have herein succeeded a new class of BioAIEgens from the abundant and low-cost natural rosin via facile Schiff base reactions." The related discussions have been involved in the revised Discussion.

11. Please indicate which statistical analysis method was employed in the biological analysis.

Responses: Thanks for your kind suggestion. We added statistical analysis method information in updated manuscript.

Statistical analysis: The values reported are expressed as mean standard deviation (SD). The Origin 8 software was used for graph plotting. Each experiment includes five replicates.

REVIEWERS' COMMENTS

Reviewer #1 (Remarks to the Author):

The authors have addressed my comments carefully and correctly. It can be accepted now.

Reviewer #2 (Remarks to the Author):

The reviewers' concerns has been addressed by the authors with additional experimental results and discussion.

Responses to reviewers' comments for the manuscript titled “**BioAIEgens Derived from Rosin: How Does Molecular Motion Affect Their Photophysical Processes in Solid State?**”

Reviewer #1 (Remarks to the Author):

The authors have addressed my comments carefully and correctly. It can be accepted now.

Responses: We sincerely thank the reviewer's appreciation and support of publication of our work.

Reviewer #2 (Remarks to the Author):

The reviewers' concerns has been addressed by the authors with additional experimental results and discussion.

Responses: We sincerely thank the reviewer for the recognition and supporting publication of our work.